# Oxygen Adsorption Induced Superconductivity in Ultrathin FeTe Film on SrTiO_3_(001)

**DOI:** 10.3390/ma14164584

**Published:** 2021-08-15

**Authors:** Wei Ren, Hao Ru, Kun Peng, Huifang Li, Shuai Lu, Aixi Chen, Pengdong Wang, Xinwei Fang, Zhiyun Li, Rong Huang, Li Wang, Yihua Wang, Fangsen Li

**Affiliations:** 1School of Nano-Tech and Nano-Bionics, University of Science and Technology of China, Hefei 230026, China; wren2019@sinano.ac.cn; 2Department of Physics and State Key Laboratory of Surface Physics, Fudan University, Shanghai 200438, China; hru16@fudan.edu.cn; 3Vacuum Interconnected Nanotech Workstation (Nano-X), Suzhou Institute of Nano-Tech and Nano-Bionics (SINANO), Chinese Academy of Sciences (CAS), Suzhou 215123, China; 18752018528@163.com (K.P.); 51171213005@stu.ecnu.edu.cn (H.L.); slu2018@sinano.ac.cn (S.L.); axchen2017@sinano.ac.cn (A.C.); pdwang2020@sinano.ac.cn (P.W.); xwfang2020@sinano.ac.cn (X.F.); zhiyunli2018@sinano.ac.cn (Z.L.); rhuang2009@sinano.ac.cn (R.H.); 4Shanghai Research Center for Quantum Sciences, Shanghai 201315, China

**Keywords:** FeTe, oxygen incorporation, superconductivity, microscopic origin

## Abstract

The phenomenon of oxygen incorporation-induced superconductivity in iron telluride (Fe_1+y_Te, with antiferromagnetic (AFM) orders) is intriguing and quite different from the case of FeSe. Until now, the microscopic origin of the induced superconductivity and the role of oxygen are far from clear. Here, by combining in situ scanning tunneling microscopy/spectroscopy (STM/STS) and X-ray photoemission spectroscopy (XPS) on oxygenated FeTe, we found physically adsorbed O_2_ molecules crystallized into *c* (2/3 × 2) structure as an oxygen overlayer at low temperature, which was vital for superconductivity. The O_2_ overlayer were not epitaxial on the FeTe lattice, which implied weak O_2_ –FeTe interaction but strong molecular interactions. The energy shift observed in the STS and XPS measurements indicated a hole doping effect from the O_2_ overlayer to the FeTe layer, leading to a superconducting gap of 4.5 meV opened across the Fermi level. Our direct microscopic probe clarified the role of oxygen on FeTe and emphasized the importance of charge transfer effect to induce superconductivity in iron-chalcogenide thin films.

## 1. Introduction

Iron-chalcogenide Fe (Se,Te) superconductors are an important family of (*T*_c_) iron-based high transition temperature (*T*_c_) superconductors. The monolayer grown on SrTiO_3_ (STO) still holds the record in *T*_c_ with much-enhanced superconducting pairing [1,2,3]; the strong spin-orbit coupling in Te-alloyed compounds leads to a topological band structure [4,5,6,7], the family also exhibits rich magnetic phases strongly dependent on the chemical composition [8,9]. Non-superconducting FeTe with antiferromagnetic (AFM) orders was considered as the parent compound of Iron-chalcogenide superconductors [10]. Superconductivity could be induced in FeTe through Se or S substitution [10] and oxygen incorporation by low-temperature annealing or long-time exposure in an O_2_ atmosphere [11,12,13,14,15,16,17,18,19,20,21]. Despite lots of efforts towards understanding oxygenated FeTe bulk crystals [15,16,17,18,19,20,21,22] and thin films [11,12,13,14,21], the microscopic role played by oxygen is still unknown.

Zheng et al. [12] found that interstitial oxygen, rather than oxygen substitution [23], was responsible for the emergence of superconductivity. The interstitial oxygen [13,16] would dope holes into FeTe [17], suppress the AFM ordering and then induce superconductivity. Yamazaki et al. [19] and Sun et al. [18] have studied the dynamics of oxygen annealing and concluded that the superconductivity first emerged on the surface and the superconducting regions moved inside with non-superconducting materials left on the surface, such as Fe_2_O_3_, TeO_x_, FeTe_2_. The controversial results from different experimental probes call for a unified study of the crystal, chemical and electronic structures on thin films with controlled oxygen incorporation. However, this is generally challenging to achieve for ex situ measurements because samples exposed to the atmosphere inevitably change surface morphology and chemistry. In this study, we overcame this challenge by integrating one ultra-high vacuum environment for the sample growth and oxygenation process along with spectroscopic tools with elemental and spatial resolution.

First, we grew nearly stoichiometric 10 Unit-Cell (UC) FeTe epitaxy ultrathin films by molecular beam epitaxy (MBE) and then studied the adsorption of O_2_ incorporation by combined in situ STM and X-ray photoelectron spectroscopy (XPS). We found that O_2_ molecules physically adsorbed on FeTe surface under low O_2_ partial pressure at room temperature and crystallized into *c* (2/3 × 2) structure, which gradually disappeared after heating up to 100 °C in vacuum. No surface oxidation can be found. The band shift observed in STS and XPS suggests hole doping from the O_2_ overlayer to FeTe, which induced a superconducting gap in FeTe. It agrees with the ex situ transport measurement that long-time exposure to air can induce superconducting transition in 10UC FeTe on STO.

## 2. Materials and Methods

High-quality 10UC FeTe thin films were grown on SrTiO_3_(001) by using a Unisoku ultrahigh vacuum (UHV) low-temperature STM system (Unisoku 1300 9T-2T-2T, 2016, Hirakata, Osaka, Japan), equipped with an MBE chamber for film preparation. The base pressure of our MBE chamber is 1 × 10^−10^ mbar and does not exceed 5 × 10^−10^ mbar during FeTe epitaxial growth. The Nb doped-STO substrate (Nb: 0.7% wt, KMT comp.) was firstly annealed at 1000 °C for 1 h and then kept at 280 °C during FeTe growth. High purity Fe (99.9999%) and Te (99.9999%) sources were co-evaporated onto SrTiO_3_(001) substrates from two standard Knudsen cells. To improve the quality of crystallization, FeTe films were annealed at 280 °C for 1 h after growth. The O_2_ gas (99.999%) was introduced to a small preparation chamber with a base pressure of 2 × 10^−9^ mbar. Then the oxygen absorption was carried out at room temperature under O_2_ partial pressure of 1.6 × 10^−4^ mbar (if not specified otherwise). After O_2_ absorption, the samples were in situ transferred to the STM chamber or XPS equipment for further characterizations. For electronic transport measurement, 10UC FeTe films were grown on high-resistance STO and covered with the Te protection layer.

All STM/STS measurements were performed at a temperature of 4.7 K with a bias voltage applied to the sample. Polycrystalline Pt/Ir tips were cleaned by electron bombardment and then carefully calibrated on Ag films grown on Si (111). All STM topographic images were obtained under a constant current mode. The differential conductance d*I*/d*V* spectra were acquired by using a standard lock-in technique at a frequency of 983 Hz. The STM topographic images were processed with WSXM software (WSxM 5.0 Develop 10.0, 2020, Julio Gómez Herrero & José María Gómez Rodríguez, Madrid, Spain) [24].

After STM/STS measurements, the samples were transferred via a UHV interconnection pipeline in Nano-X for XPS/UPS characterization without exposure to air. XPS/UPS experiments were carried out by using a commercial analyzer (PHI 5000, VersaProbe, ULVAC-PHI, ~ 5 × 10^−10^ mbar, 2016, Chigasaki, Kanagawa, Japan), equipped with a monochromatic Al K_α_ X-ray source of 1486.7 eV. The binding energy (BE) of core-level peaks were calibrated with respect to the C-C 1s bond (BE = 284.8 eV). A monochromatized He I with an energy of 21.22 eV were used for UPS measurement. During the UPS measurement, a voltage of −5 V was applied between the sample and the spectrometer.

## 3. Results

High-quality FeTe films are epitaxially grown on STO substrates via step flow growth mode. Figure 1a shows the typical surface morphology of 10UC FeTe film, and Figure 1b shows the atomically resolved image with Te termination. Fast Fourier transform image yields enlarged lattice constant of ~ 0.384 nm, similar to the case of FeSe on STO [1]. No Fe adatoms (or interstitial Fe atoms) and Te vacancies can be observed on the surface, which indicates nearly stoichiometric FeTe epitaxy films. To carry out electronic transport measurement, the protection Te layer was deposited on the 10UC FeTe surface, and the results were shown in Figure 1c. No superconducting transition but semiconducting behavior was observed in initially grown 10UC FeTe film down to 1.8 K, consistent with previous experiments [25,26]. Compared with bulk or thick FeTe films [11,13,23,27,28,29], we have not found an anomaly in the *R*-*T* curve between 40 K and 80 K, which further demonstrates free of Fe interstitial atoms [26]. AFM state may be suppressed here. After exposing the sample to air for 8 days, the resistance turned downward at ~3.2 K, as shown in the inset of Figure 1c. Considering the fact of long-time exposure-induced superconductivity in FeTe compounds, such downward should be attributed to superconducting transition. It is further verified by the suppression of transition under external magnetic fields in Figure 1d, characteristic of superconductivity. Non-zero resistance here suggests the developed superconductivity is not uniform. We found that exposure to air for a longer time would increase the superconducting transition temperature further.

To figure out the superconducting mechanism in FeTe after exposure to O_2_, we carried out the oxygenated process under a well-control condition and probed the atomic and electronic structures microscopically. Figure 2a shows the surface morphology of the initial adsorption stage after exposure 10UC FeTe to O_2_ under a partial pressure of 1.6 × 10^−4^ mbar for 15 min. We can observe extra layers along the step edge, while other uncovered clean FeTe areas still remain unchanged, as illustrated by the atomically resolved STM image in the inset. No O_Te_ substitution or interstitial O atoms can be observed. The enlarged image in Figure 2b shows there are small compacted islands with a height of ~0.18 nm in the overlayer, as shown in Figure 2c. Some dispersed clusters can be also found at the domain boundaries, which should be the ones that have not coarsened into the islands. If further increasing the amount of inlet O_2_, nearly the whole surface can be covered by the adsorbed overlayers, as shown in Appendix A. The uncovered areas are still very clean in Appendix A.

To unravel the nature of adsorbed overlayer (O_2_ molecules, FeO_x_, TeO_x_, Fe-Te-O or others?), Figure 2d,e present the core-level XPS spectra of Te 3d and Fe 2p before and after inlet O_2_, respectively. After inlet O_2_ under a partial pressure of 1.6 × 10^−4^ mbar, we do not observe Te-O and Fe-O spectra signals. If exposure to O_2_ partial pressure of 1.5 × 10^−1^ mbar for 20 min, as shown in Appendix A clear Te-O core level peak can be identified, which suggests surface oxidation under high O_2_ partial pressure. The XPS spectra of Te 3d and Fe 2p show an obvious peak shift toward higher binding energy, which hints hole doping effect in FeTe. The overlayer would gradually desorb when heating up to 100 °C and disappeared after annealing at 200 °C for 2 h. Only a clean FeTe surface was left. The overlayer will re-adsorb after inlet O_2_ again. Thus, we can rule out the possibility of Fe- or Te-related compounds because there was negligible residual Fe or Te atoms after growth. In addition, we notice there are some dispersed clusters higher than the compacted islands in Figure 2b and Figure 3a, which helps us to conclude that the overlayer should be O_2_ molecules rather than stacked O atoms. The shape of O 1s core level almost remains unchanged after O_2_ adsorption under a partial pressure of 1.6 × 10^−4^ mbar, as illustrated in Appendix A, but changes a lot after surface oxidation due to the signal from O-Te and O-Fe. It implies the signal of O 1s at 529.5 eV ~ 531 eV are mainly from the STO substrate, which, as expected, exhibits a similar core level shift as Te 3d, as shown in Appendix A. It also consists of the nature of physically adsorbed O_2_ molecules, where characteristic core-level XPS peak always cannot be found. Thus, we name the overlayer heterostructure as FeTe : O_2_. UPS measurements in Figure 2f show that the work function just slightly changed from *ϕ*_FeTe_ ~ 4.43 eV to *ϕ*_FeTe:O_2__ ~ 4.38 eV after O_2_ adsorption. The small difference in work function is also consistent with the nature of O_2_ molecules in the overlayer.

Figure 3a shows some small compacted FeTe : O_2_ islands, where striped structures can be clearly observed. Only two perpendicular orientations were distinguished, which consist of the tetragonal symmetry of the underlying FeTe lattice. The atomically resolved structure of adsorbed FeTe : O_2_ islands is shown in Figure 3b. The distance between neighboring bright rows is ~ 0.386 nm, nearly equal to the lattice constant *a*_FeTe_ of FeTe. The corresponding fast Fourier transform image in Figure 3c yields an in-plane lattice of *n*_1_ with a value of 0.25±0.02 nm and *n*_2_ with a value of 0.41±0.02 nm. The angle between *n*_1_ and *n*_2_ is measured to be ~ 72°. We notice that the spot distance of 0.25 nm along *n*_1_ is nearly equal to 2/3*a*_FeTe_. Thus, the lattice can be notated as *c* (2/3×2), as shown in Figure 3b,c. Such lattice arrangement demonstrates a close relationship between the O_2_ adsorption overlayer (FeTe : O_2_) and the FeTe surface. To locate the precise adsorption sites of O_2_ on FeTe, we achieved an atomically resolved image on both the FeTe : O_2_ island and FeTe surface in Figure 3d. The results are shown in the structural model in Figure 3e. As illustrated, along the *n*_1_ direction, the adsorption sites of O_2_ molecules are not well-defined relative to FeTe lattice and not at high-symmetry points. It not only signifies that the van der Waals interaction between O_2_ molecules plays a vital role in the overlayer but also verifies the absence of chemical bonding between the O_2_ interlayer and the FeTe surface.

To explore how the electronic structure changes due to O_2_ absorption, we conducted detailed differential conductance d*I*/d*V* measurements, which are known to be proportional to the local density of state (LDOS). On as-prepared and uncovered clean 10UC FeTe surfaces, the STS remain nearly unchanged with a “V” shape across the Fermi level in Figure 4a, consistent with previous studies [2,30]. A hump at the negative bias voltage shift from −0.37 V to −0.325 V on FeTe : O_2_ after O_2_ adsorption, which indicates hole doping effect to FeTe, agreeing well with the observation of peak shift in XPS experiments in Figure 2. Such oxygen-induced hole doping effect [31] was recently observed on the CsV_3_Sb_5_ with an increased superconducting transition temperature. Additionally, hole doping effects were observed in Bi_2_Te_3_/FeTe [32,33] and Sb_2_Te_3_/FeTe [26] heterostructures and thought to be the origin of two-dimensional superconductivity in the heterostructures. Figure 4b shows the d*I*/d*V* spectra in a small energy range. A clear V-shape was observed on a clean FeTe surface, similar to the one on the monolayer FeTe [2] and thick FeTe film [30]. Strikingly, on FeTe : O_2_ islands two pronounced peaks with an energy gap of ~9 meV were clearly identified across the Fermi level. Considering the facts of previous observation of hole-doping-induced superconductivity in the Bi_2_Te_3_/FeTe heterostructure and the superconductivity observed in 10UC FeTe/STO sample after long-time exposure to air, we conclude that the gap on FeTe : O_2_ is just a superconducting gap with *Δ* ~4.5 meV, larger than the one on Bi_2_Te_3_/FeTe heterostructure (~2.5 meV) [32,34]. It should be due to the interfacial effect on STO substrate because similar enhanced superconductivities have been observed on the Pb islands [35] and monolayer FeTe_x_Se_1-x_ [1,2,25]. We have studied the spatial distribution of the superconducting gap on the FeTe : O_2_ islands in Figure 4c. Pronounced coherent peaks can be observed in all the d*I*/d*V* curves.

The observation of enhanced superconducting gap on FeTe : O_2_ islands is intriguing and distinct from all the previously proposed mechanisms. Physically adsorbed O_2_ molecules rather than interstitial oxygen or oxygen substitution are responsible for superconductivity in oxygenated FeTe. Previously, annealing under an elevated temperature or quite a long-time aging may be aimed at the formation of a uniform O_2_ overlayer into the FeTe layer. Our study here also emphasizes the importance of the hole doping effect and interfacial effect to induce superconductivity on FeTe. We have not observed a superconducting gap on FeTe:O_2_ overlayer on monolayer FeTe, which may be due to the extra electron doping from the STO substrate.

## 4. Conclusions

In summary, we have systematically investigated the changes of morphology and electronic structure of 10UC FeTe films after O_2_ adsorption. E*x*-*situ* transport measurements demonstrate long-term exposure to air could induce superconducting transition in 10UC FeTe ultrathin film, which verifies the effect of oxygen incorporation. Initially, O_2_ molecules physically adsorbed on the FeTe surface and formed compacted FeTe : O_2_ islands with c(2/3×2) structure at low temperatures. No oxygen-related defects can be detected on the uncovered clean FeTe surface. Energy shifts in STS and XPS measurements point out obvious hole doping into FeTe from the oxygen overlayer. Such hole doping results in a superconducting gap of ~ 4.5 meV opened with apparent coherent peaks. Our study here directly provides microscopical insights about the role of oxygen on FeTe thin film.

## Figures and Tables

**Figure 1 materials-14-04584-f001:**
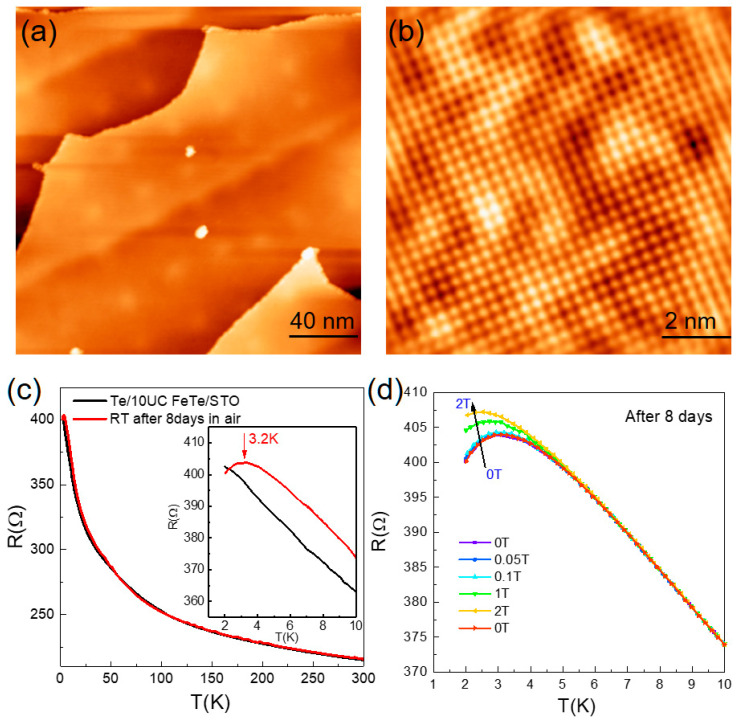
(Color online) (**a**) Surface topography (200 nm × 200 nm) of the 10UC FeTe films on the Nb:STO. Scanning condition: sample bias 𝑉_s_ = 3 V, tunneling current 𝐼_𝑡_ = 100 pA. (**b**) Typical atomically resolved STM image (10 nm ×10 nm) of 10UC FeTe. Scanning condition: sample bias 𝑉_s_ = 0.7 V, tunneling current 𝐼_𝑡_ = 200 pA. (**c**) Temperature-dependent resistance (*R-T*) curve of 10UC FeTe thin film on HR-STO before and after exposure to air for 8 days. Inset: *R-T* curve under low temperature, showing turn-down behavior after exposure to air. (**d**) Temperature-dependent resistance of 10UC FeTe after exposure to air for 8 days under an out-of-plane magnetic field up to 2 T. The turn-down behavior was suppressed under magnetic fields and recovered after magnetic field withdrawal.

**Figure 2 materials-14-04584-f002:**
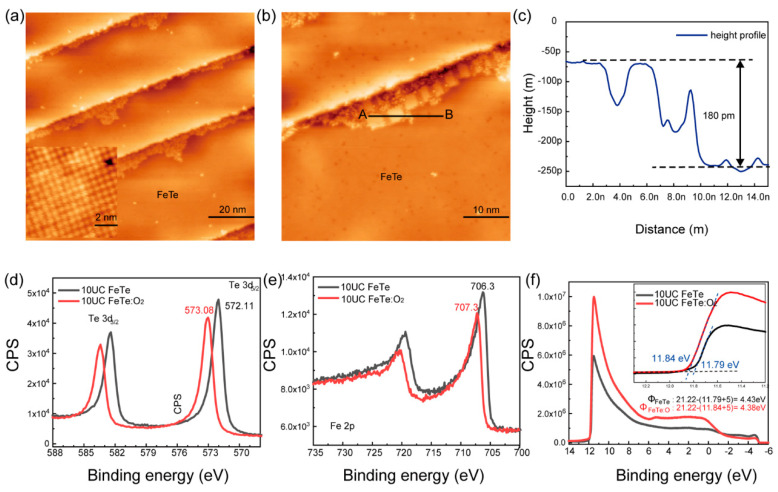
(Color online) (**a**) Surface topography (100 nm × 100 nm) of the 10UC FeTe films after exposure to oxygen partial pressure of 1.6 × 10^−4^ mbar for 15 min at RT. Scanning condition: sample bias 𝑉_s_ = 1 V, tunneling current 𝐼_𝑡_ = 100 pA. O_2_ overlayers were initially formed along the step edge. Inset is the atomically resolved STM image of the uncovered clean FeTe surface. (**b**) Enlarged STM image (50 nm × 50 nm) of O_2_ overlayers (FeTe : O_2_) on FeTe surface. Scanning condition: sample bias 𝑉_s_ = 0.50 V, tunneling current 𝐼_𝑡_ = 100 pA. (**c**) The height profile of FeTe : O_2_ along line AB in panel b. (**d**,**e**) XPS core-level spectra of Te 3d (**d**) and Fe 2p (**e**) on as-prepared 10 UC FeTe and FeTe : O_2_ covered surface, respectively. No trance of Te-O and Fe-O signals can be observed. The spectra show apparent peak shifts after oxygen adsorption. (**f**) UPS spectra of 10UC FeTe covered *w*/*o* FeTe : O_2_, and the added voltage here is −5 V. Inset is the large view.

**Figure 3 materials-14-04584-f003:**
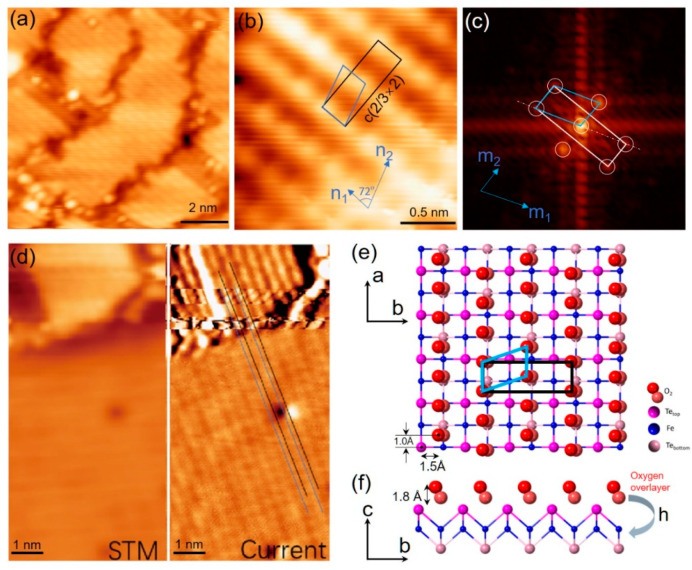
(Color online) (**a**) Small compacted FeTe : O_2_ islands on 10UC FeTe. Dispersed clusters can be observed at domain boundaries. Scanning condition: sample bias 𝑉_s_ = 1 V, tunneling current 𝐼_𝑡_ = 200 pA. (**b**) Atomically resolved STM images (2 nm × 2 nm) on FeTe : O_2_ islands. Scanning condition: sample bias 𝑉_s_ = 0.1 V, tunneling current 𝐼_𝑡_ = 200 pA. (**c**) The fast Fourier transform image of (**b**). White circles mark the reciprocal lattice. (**d**) STM topographic (left panel) and constant current (right panel) images (5 nm × 10 nm) of the 10UC FeTe films with one FeTe : O_2_ island. The current image helps us to measure the adsorption geometrical configuration. Scanning condition: sample bias 𝑉_s_ = 0.3 V, tunneling current 𝐼_𝑡_ = 300 pA. (**e**,**f**) Schematic diagrams of geometrical configuration of O_2_ adsorption overlayer on FeTe: (**e**) top side and (**f**) side views.

**Figure 4 materials-14-04584-f004:**
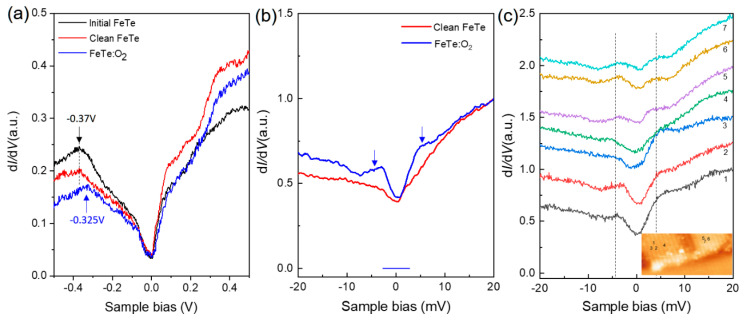
(Color online) (**a**) Typical d*I*/d*V* spectra on 10UC FeTe before and after O_2_ adsorption. A hump at negative energy shows a clear shift on FeTe : O_2_ overlayer. Starting condition: 𝑉_s_ = 0.5 V, tunneling current 𝐼_𝑡_ = 100 pA. (**b**) The zoomed-in d*I*/d*V* spectra on clean FeTe area and FeTe : O_2_ overlayer. An energy gap with two pronounced peaks was observed across the Fermi level, which is a superconducting gap. Starting condition: 𝑉_s_ = 20 mV, tunneling current 𝐼_𝑡_ = 200 pA. (**c**) A series of d*I*/d*V* spectra on FeTe : O_2_, showing the energy gap can be observed on the whole FeTe : O_2_ islands. Starting condition: 𝑉_s_ = 20 mV, tunneling current 𝐼_𝑡_ = 200 pA.

## Data Availability

The data presented in this study are available on request from the corresponding authors.

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
