# Peer review of "Oxygen Adsorption Induced Superconductivity in Ultrathin FeTe Film on SrTiO3(001)"

_materials, 2021, doi:10.3390/ma14164584_

Round 1

Reviewer 1 Report

The work "Oxygen adsorption induced superconductivity in ultrathin FeTe film on SrTiO3 (001)" is very interesting and well written, all aspects of the experiment and research part of the work are presented. The conclusions correspond to the tasks set. The oxygen-induced superconductivity in iron telluride 16 (Fe1+yTe, with antiferromagnetic (AFM) order) noted in this work is of great interest to the readers of the rapidly developing direction of non-traditional superconductivity.

In addition, I want to highlight one necessary addition that needs to be made in the Supplementary Material. Figure S2 shows three peaks in the carbon region. Please explain how you eliminate the contribution of adsorbed carbon to the surface? The XPS spectrum shows about three peaks, probably carbon oxides.

As the authors of the work note, the phenomenon of oxygen-induced superconductivity in iron telluride (Fe1 + yTe, with AFM order) is intriguing and differs from the case of FeSe. Important in this work are the microscopic nature of induced superconductivity and the role of oxygen on the surface of compounds with FeTe. The authors here qualitatively investigated the results of the formation of local oxygen centers. This is important because the surface is always oxidized in air. To use it in products, you need to take this into account. In-situ scanning tunneling microscopy combined with XPS is a more accurate and detailed technique. As the authors note, the O2 layer was not epitaxial on the FeTe lattice, which implies a weak O2-FeTe interaction. A very important aspect is that the authors noted in the measurements on the effect of hole doping from the upper O2 layer to the FeTe layer, which leads to the appearance of a superconducting gap of 4.5 meV, open across the Fermi level. The authors explained the role of oxygen in FeTe and emphasized the importance of the charge transfer effect for superconductivity in thin films of iron chalcogenides. The only thing in the work that I was interested in was the contribution from adsorbed carbon on the FeTe surface together with oxygen. This is an additional challenge. The authors in the work noted the most important thing is that hole doping from adsorbed oxygen contributes to superconducting properties.

Reviewer 2 Report

The paper presents new results for oxygen absorption-induced superconductivity in FeTe films on STO substrate. There are some shortcomings which must be improved prior to publication. 

Language corrections required. Advice of a professional service recommended. 

Add list of abbreviations, e.g., for STO, STM, STS, STS, XPS, UPS, MBE, etc. 

Materials and methods section is too short and needs expansion. In particular, regarding MBE, O2 absorption (gas purity), STM, etc.

line 102: I do not understand "AFM state may be suppressed"

line 115: HR-STO ?

line 144: do not use "haven't"
